# LSTMEMBED: a Lexical and SemanTic Model of Embeddings with a bidirectional LSTM

## Abstract

The Long Short-Term Memory (LSTM) architecture for recurrent neural networks has become the state-of-the-art model for a range of different Natural Language Processing (NLP) tasks, especially in language modeling and sequence to sequence learning. In this paper we leverage a bidirectional LSTM while at the same time taking advantage of other semantic resources in order to create a vector space model for words and senses that outperforms most popular algorithms for learning embeddings. We evaluate our approach on the most well-known benchmarks on vector space representations.

## 1 Introduction

Recurrent neural networks (RNNs) with Long Short-Term Memory (LSTMs) have recently gained considerable popularity. Introduced by Hochreiter and Schmidhuber (1997), LSTMs are a special kind of RNN capable of learning long-term dependencies on problems related with sequential data. RNNs, and particularly LSTMs have been working extremely well on a large variety of problems in NLP, such as machine translation (Cho et al., 2014), lexical substitution (Melamud et al., 2016), word sense disambiguation, (Kågebäck and Salomonsson, 2016; Yuan et al., 2016), syntactic parsing (Dyer et al., 2015), among others.

Embeddings represent lexical and semantic items in a low-dimensional continuous space. The resulting vectors capture useful syntactic and semantic information, such as regularities in language, where relationships are characterized by a relation-specific vector offset. Recent approaches, such as word2vec (Mikolov et al., 2013), and GloVe (Pennington et al., 2014) are efficient for learning embeddings, but they do not take into account word ordering. Vice versa, RNNs take order into account but they are not competitive in terms of speed or quality (Mikolov et al., 2010; Mikolov and Zweig, 2012; Mesnil et al., 2013).

The recently celebrated LSTMs appear to be perfect for learning sequence representations, like phrases (Hill et al., 2016) and contexts (Melamud et al., 2016). However, when dealing with large vocabularies, LSTMs involve time-intensive matrix-matrix multiplications, making them prohibitively expensive. This issue was addressed by Jean et al. (2015), who proposed an approximate training algorithm based on sampling, and Zoph et al. (2016), who introduced an adaptation for GPUs of the noise-contrastive estimation algorithm, a method that avoids repeated summations by training the model to correctly separate generated noise samples from words observed in the training data. However, both these approaches speed up the training process at the cost of lowering the performance.

Our contributions are twofold:

- We introduce LSTMEmbed, an RNN model based on a bidirectional LSTM for training word and sense embeddings that which outperforms classical approaches such as word2vec and GloVe.

- We present an innovative idea for enriching these representations with semantic knowledge from large corpora and vocabularies, while at the same time speeding up the training process.

## 2 Vector Space Models for words and senses

### 2.1 Word Embeddings

Embeddings represent lexical and semantic items as real-valued continuous vectors. These representations are extremely good at capturing syntactic and semantic regularities in language as well as relationships among items. Recent advances (Mikolov et al., 2013, word2vec) showed that word representations learnt with a neural network trained with raw text show relationships such as the male/female relationship, e.g. the induced vector representations of $king - man + woman$ resulted very close to the induced vector of $queen$. GloVe, an alternative approach trained on aggregated global word-word co-occurrences, reached similar results. While these embeddings are surprisingly good for monosemous words, they fail to represent properly the non-dominant senses of words. For instance, the representations of $bar$ and $pub$ should be similar, as well as those of $bar$ and $stick$, but having similar representations of $pub$ and $stick$ is undesired. Several approaches were proposed to address this problem: Yu and Dredze (2014) presented an alternative way to train word embeddings by using, in addition to common features, words having some relation in a semantic resource, like PPDB[1] (Ganitkevitch et al., 2013) or WordNet (Miller, 1995). Faruqui et al. (2015) presented a technique applicable to pre-processed embeddings, in which vectors are updated (i.e. retrofitted) in order to make them more similar to those which share a word type and less similar those which do not. The word types were extracted from diverse semantic resources such as PPDB, WordNet and FrameNet (Baker et al., 1998). Finally, Melamud et al. (2016) introduced context2vec, a model based on a bidirectional LSTM for learning word embeddings. They use large raw text corpora to learn a neural model that embeds entire sentential contexts and target words in the same low-dimensional space.

### 2.2 Sense Embeddings

In contrast with the above approaches, which aim to learn representations of single words, sense embeddings represent individual word senses as separate vectors. Sense embeddings can learned with the main approaches: (1) supervised, which rely on a predefined sense inventory such as Word-Net, Wikipedia, BabelNet[2] (Navigli and Ponzetto, 2012) or Freebase (Bollacker et al., 2008), and (2) unsupervised, which rely on contextual information from parallel corpora to discriminate word senses among different occurrences. From the first group we can highlight successful approaches: SENSEMBED (Iacobacci et al., 2015) used Babelfy[3], a state-of-the-art tool for Word Sense Disambiguation and Entity linking, to build a sense-annotated corpus which was in turn used to train a vector space model for word senses with word2vec. SENSEMBED exploits the structured knowledge of BabelNet's sense inventory along with the distributional information gathered from text corpora. An important limitation of this approach was the inability to train both word and sense embeddings. The only word embeddings that the model was able to learn were the representations of words which were not annotated by the disambiguation tool. Those representations were of poor quality due to the fact that they are learnt from the occurrences in a ambiguous or unclear context, else the disambiguation should be able to annotate them. In addition, owing to the fact that it is based on word2vec, this approach suffers from the lack of word ordering. AutoExtend (Rothe and Schütze, 2015) is a system that learns embeddings for lexemes, senses and synsets from WordNet in a shared space. The synset/lexeme embeddings live in the same vector space as the word embeddings, given the constraint that words are sums of their lexemes and synsets are sums of their lexemes. AutoExtend is based on an autoencoder, a network that mimics the input and output vectors. Finally, Mancini et al. (2016) presented SW2V, an extension of word2vec which learns word and sense embeddings jointly and represents them in a unified vector space. The model was built by exploiting large corpora and knowledge obtained from WordNet and BabelNet. Their basic idea was to extend the CBOW architecture of word2vec to represent both words and senses as different inputs and train the model in order to predict the word and the sense in the middle. In contrast to AutoExtend, the model learns word and sense embeddings in a shared space as an emerging feature, rather than via constraints on both representations. Nevertheless, being based also

---

[1]www.paraphrase.org/#/download

[2]http://www.babelnet.org/
[3]http://babelfy.org/

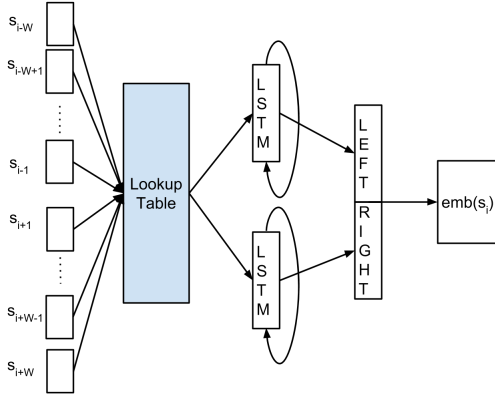

Figure 1: The LSTMEmbed architecture

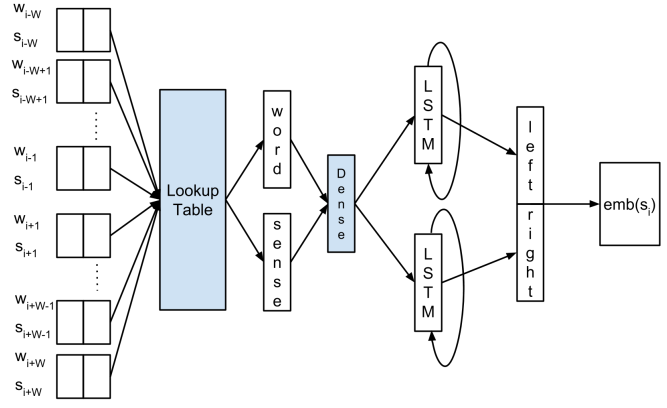

Figure 2: The LSTMEmbed$_{SW}$ extension

on word2vec, SW2V also lacks word order. In marked contrast, LSTMEmbed aims to learn representations for both words and senses in a shared emerging space, handling word ordering, and improving those representations by injecting semantic knowledge via pretrained embeddings.

## 3 LSTMEmbed

### 3.1 Model Overview

At the core of LSTMEmbed is a bidirectional LSTM, a special kind of recurrent neural network (RNN). An RNN is a type of neural network architecture particularly suited for learning time series. The simplest RNN is called Elman network. This network is composed of an input, a hidden and an output layer, just as regular feed-forward networks, with the difference that the hidden layer takes in addition to the input vector $\mathbf{x}_t \in \mathbb{R}^n$, its previous state $\mathbf{h}_{t-1} \in \mathbb{R}^m$. Mathematically, the computation follows the formula:

$$\mathbf{h}_t = f\left(\mathbf{W}\mathbf{x}_t + \mathbf{U}\mathbf{h}_{t-1} + \mathbf{b}\right) \quad (1)$$

where $\mathbf{W} \in \mathbb{R}^{m \times n}$ is the weights matrix from the input to the hidden layer, $\mathbf{U} \in \mathbb{R}^{m \times m}$ is the weights matrix from the previous hidden state to the current one, $\mathbf{b} \in \mathbb{R}^m$ is the bias and $f$ is the activation function. In practice this architecture has several problems in learning long-term dependencies (Hochreiter, 1991; Bengio et al., 1994). To cope the latter, we adopt the Long Short Term Memory (LSTM) (Hochreiter and Schmidhuber, 1997) architecture which augments the RNN with a series of connections called gates: three gates are added namely the input ($i_t$), the forget ($f_t$) and the output ($o_t$) gates. These gates control the flow of information between states. This is achieved

by adding an extra vector, $\mathbf{c}_t \in \mathbb{R}^n$: in each step, the LSTM takes $\mathbf{x}_t$, $\mathbf{h}_{t-1}$ and $\mathbf{c}_{t-1}$ as input and outputs $\mathbf{h}_t$ and $\mathbf{c}_t$ according to the following calculations:

$$\begin{aligned}
\mathbf{i}_t &= \sigma\left(\mathbf{W}^i\mathbf{x}_t + \mathbf{U}^i\mathbf{h}_{t-1} + \mathbf{b}^i\right) \\
\mathbf{f}_t &= \sigma\left(\mathbf{W}^f\mathbf{x}_t + \mathbf{U}^f\mathbf{h}_{t-1} + \mathbf{b}^f\right) \\
\mathbf{o}_t &= \sigma\left(\mathbf{W}^o\mathbf{x}_t + \mathbf{U}^o\mathbf{h}_{t-1} + \mathbf{b}^o\right) \quad (2) \\
\mathbf{g}_t &= tanh\left(\mathbf{W}^g\mathbf{x}_t + \mathbf{U}^g\mathbf{h}_{t-1} + \mathbf{b}^g\right) \\
\mathbf{c}_t &= \mathbf{f}_t \odot \mathbf{c}_{t-1} + \mathbf{i}_t \odot \mathbf{g}_t \\
\mathbf{h}_t &= \mathbf{o}_t \odot tanh(\mathbf{c}_t)
\end{aligned}$$

where $\sigma$ is the sigmoid function, $tanh$ is the hyperbolic tangent and $\odot$ is the element-wise multiplication operator. The bidirectional LSTM (Graves and Schmidhuber, 2005, BLSTM) is a variant of the original LSTM especially designed for temporal problems where the input direction is relevant. The state at each time step in a BLSTM consists of the state of two LSTMs, one going left and one going right.

LSTMEmbed, however, is not merely a standard BLSTM: it produces word and sense embeddings jointly based on the following innovations:

- A single lookup table, shared between both left and right LSTM, learned in the same architecture that represents both words and senses.

- Using a sense-annotated corpus which includes both words and senses for learning the embeddings.

- A new way of learning using a set of pretrained embeddings as the objective for injecting semantic information while speeding up the training.

### 3.2 Definition

As we mentioned before, our model uses a bidirectional LSTM to obtain word and sense representations. Given a sense-annotated corpus and a set of pretrained embeddings, the objective is to predict the embedding of a single word or sense (embedding given by the pretrained set) given its context. The context is defined by a fixed window $W$ of words and senses on each side.

Figure 1 illustrates the model architecture. Given a sentence and the position of the target token $i$ (word or sense), the input to our model are two sequences of tokens $s_{i-W}, ..., s_{i-1}$, the preceding context, and $s_{i+1}, ..., s_{i+W}$, the posterior context. Each token is represented by its corresponding embedding vector $\mathbf{v}(s_j) \in \mathbb{R}^n$, given by a shared table which allows to learn representations taking into account the contextual information on both sides of the sentence. Then, the BLSTM reads both sequences, the LSTM that reads the preceding context from left to right and the LSTM that reads the posterior context, from right to left:

$$
\begin{aligned}
o_l &= lstm_l(\mathbf{v}(s_{i-W}), ..., \mathbf{v}(s_{i-1})) \\
o_r &= lstm_r(\mathbf{v}(s_{i+1}), ..., \mathbf{v}(s_{i+W}))
\end{aligned}
\quad (3)
$$

The output of the model is given by a perceptron with sigmoid as its activation function, applied to the concatenation of the outputs of both LSTMs:

$$
out_{LSTMEmbed} = \sigma\left(\mathbf{W}^o(o_l \oplus o_r)\right) \quad (4)
$$

where $\mathbf{W}^o \in \mathbb{R}^{2m \times m}$ is the weights matrix and $m$ the dimension of the LSTM. Then, the model compares $out_{LSTMEmbed}$ with $\mathbf{emb}(s_i)$, where $\mathbf{emb}(s_i)$ is the embedding vector of the target token given by the set of pretrained embeddings. The weights of the network are modified in order to maximize the similarity between $out_{LSTMEmbed}$ and $\mathbf{emb}(s_i)$. For error calculation the comparison is in terms of cosine similarity.

$$
similarity = \mathcal{S}(\vec{v_1}, \vec{v_2}) = \frac{\vec{v_1} \cdot \vec{v_2}}{\|\vec{v_1}\|\|\vec{v_2}\|} \quad (5)
$$

Once the training is over, the embedding vector of an item $s$ that our model learns is given by $\mathbf{v}(s)$.

### 3.3 Modeling words and senses together

The vanilla version of LSTMEmbed is able to learn either word embeddings or sense embeddings. In the following we describe an extension of LSTMEmbed, LSTMEmbed$_{SW}$ (Figure 2), which learns both word and sense embeddings jointly in a shared vector space. Given a sentence and the position of the target item $i$, the input to our model are four sequences, two that represent words and senses on the left side of the context ($w_{i-W}, ..., w_{i-1}$ and $s_{i-W}, ..., s_{i-1}$) and two sequences that represent words and senses from the right side of the context ($w_{i+1}, ..., w_{i+W}$ and $s_{i+1}, ..., s_{i+W}$).

The architecture is augmented by adding an extra layer that takes the $sum$ of the embedding vectors from the word and the sense with the same index and projects them via a linear perceptron into the corresponding LSTM:

$$
p_i = \mathbf{W}\left(\mathbf{v}(s_i) + \mathbf{v}(w_i)\right) \quad (6)
$$

where $\mathbf{W} \in \mathbb{R}^{n \times n}$ is the weights matrix. In the same way as explained above, the BLSTM reads the two resulting sequences:

$$
\begin{aligned}
o_l &= lstm_l(p_{i-W}, ..., p_{i-1}) \\
o_r &= lstm_r(p_{i+1}, ..., p_{i+W})
\end{aligned}
\quad (7)
$$

### 3.4 Input alternatives & Objective embeddings

| Corpus type | Pretrained Embeddings | Representations words | senses |
|---|---|---|---|
| raw text | words | ✓ | – |
| | senses | – | – |
| sense-annotated | words | ✓ | ✓ |
| | senses | ✓ | ✓ |

Table 1: LSTMEmbed configuration alternatives.

Our model can exploit both raw and sense-annotated corpora and different types of pretrained embeddings. Given a raw-text training corpus and a set of pretrained word embeddings (i.e. word and sense embeddings), our model is only capable to learn word embeddings, while the combination of raw text and sense embeddings is invalid (see Table 1). With a sense annotated corpus the model can exploit either word or sense embeddings and also it is able to learn word or sense embeddings.

| $bar_1^n$ | $pub_1^n$ | $bar_3^n$ | $stick_1^n$ |
|---|---|---|---|
| $cafe_1^n$ | $coffee\_shop_n^2$ | $cage_n^5$ | $washer_n^2$ |
| $grill_1^n$ | $public\_bar_n^1$ | $rigid\_frame_n^1$ | $strap_n^3$ |
| $mug_4^n$ | $grill_1^n$ | $nut_n^3$ | $flap_n^1$ |
| $doughnuts_2^n$ | $bar_1^n$ | $coupler_n^1$ | $twine_n^1$ |
| $pantry_n^1$ | $wine\_bar_1^n$ | $tappet_n^1$ | $buckle_v^1$ |
| $buffet_n^2$ | $diner_1^n$ | $puley_n^1$ | $piece\_of\_leather_n^1$ |
| $coffee\_shop_n^2$ | $wallet_1^n$ | $stampings_n^1$ | $threaded_v^5$ |

Table 2: Closest senses of ambiguous nouns.

### 3.5 Inspiration from Language Models

LSTMEmbed is inspired by the architecture or recurrent neural network language models (RNN-LM) (Mikolov et al., 2010; Sutskever et al., 2011; Sundermeyer et al., 2012; Zaremba et al., 2014). Language modelling aims to learn a function which defines probability distributions over sequences of words. While RNN-LMs learns to predict target words based on their context, our model learns to predict the embedding of the target word, and uses the learning process for deriving useful representations of words and senses.

## 4 Similarity Measurement

In the following we describe how we leverage our representations for the computation of word similarity and analogy tasks.

### 4.1 Word similarity

As was claimed by Rubenstein and Goodenough (1965) the proportion of words common to the contexts of two words A and B is a function of the degree in which A and B are similar in meaning. LSTMEmbed is intended for improving word and sense representations by learning from their contexts. While measuring semantic similarity between words we will refer to the conventional approach of Resnik (1999) where the similarity bewteen two words is given by the similarity of their closest senses:

$$Sim(w_1, w_2) = \max_{\substack{s_1 \in \mathcal{S}_{w_1} \\ s_2 \in \mathcal{S}_{w_2}}} \mathcal{S}(\vec{s_1}, \vec{s_2}) \quad (8)$$

where $\mathcal{S}_{w_i}$ is the set of senses associated with the word $w_i$.

### 4.2 Word Analogy

Given two pairs of words $(w_{i1}, w_{i2})$ and $(w_{j1}, w_{j2})$, the degree of similarity between the

relation among $w_{i1}$ and $w_{i2}$, and $w_{j1}$ and $w_{j2}$ suggests by what degree the two relations are analogous. For measuring this similarity we follow Iacobacci et al. (2015):

$$An(w_{i1}, w_{i2}, , w_{j1}, w_{j2}) = \mathcal{S}(\mathbf{v}(w_{i1}) - \mathbf{v}(w_{i2}), \\ \mathbf{v}(w_{j1}) - \mathbf{v}(w_{j2})) \quad (9)$$

## 5 Experiments

### 5.1 Evaluation Methods

We evaluate our semantic representations of words and senses with different configurations on tasks of word similarity, synonym identification and word analogy.

**Word Similarity.** We evaluate LSTMEmbed on standard word similarity and relatedness datasets: the RG65 (Rubenstein and Goodenough, 1965) and the WordSim-353 (Finkelstein et al., 2002, WS353) datasets. We also include the proposed split made by Agirre and Soroa (2009) who divided WS353 in pairs that measured the degree of similarity (WSSim) and pairs that measured the degree of relatedness (WSRel). We include two large datasets: SimLex999 (Hill et al., 2015), which puts especially focus on representing antonyms as completely unrelated words, and the MEN dataset introduced by Bruni et al. (2014) composed by 3000 pairs. Two datasets especially made for analyzing verb similarity are also included in our experiments: YP130, created by Yang and Powers (2005), based on the taxonomy of WordNet, and SimVerb3500 (Gerz et al., 2016) a newer and much larger dataset extracted from the USF norms data set[4] (Nelson et al., 2004) and VerbNet[5] (Kipper et al., 2008). Finally, we include evaluations on Stanford Contextual Word Similarities (SCWS), a dataset for measuring word-in-context similarity (Huang et al., 2012). We only report the MaxSim measurement, that is, without taking into account the contextual sentences.

**Synonym Identification.** For synonym identification we include two datasets. The first one, introduced by Landauer and Dumais (1997, TOEFL), is extracted directly from the synonym questions of the TOEFL (Test of English as a Foreign Language). The test is composed by

---

[4] http://w3.usf.edu/FreeAssociation/
[5] http://verbs.colorado.edu/verb-index

| Model | WS353 | WSSim | WSRel | RG65 | MEN | SimLex | SCWS | YP130 | SimVerb | TOEFL | ESL | SemEval-2012 MaxDiff | Spearman |
|---|---|---|---|---|---|---|---|---|---|---|---|---|---|
| word2vec | 0.620 | 0.699 | 0.478 | **0.791** | 0.650 | 0.441 | 0.547 | 0.694 | 0.433 | 0.875 | **0.640** | 0.404 | 0.242 |
| GloVe | 0.575 | 0.684 | 0.426 | 0.761 | 0.675 | **0.478** | 0.532 | 0.658 | 0.399 | 0.875 | 0.620 | **0.420** | **0.265** |
| LSTMEmbed | **0.634** | **0.710** | **0.510** | 0.719 | **0.690** | 0.456 | **0.552** | **0.705** | **0.436** | **0.878** | 0.600 | 0.401 | 0.216 |
| LSTMEmbed$_{SW}$ | 0.561 | 0.626 | 0.478 | 0.785 | 0.644 | 0.450 | 0.529 | 0.600 | 0.414 | 0.869 | 0.560 | 0.358 | 0.139 |

Table 3: Spearman correlation for Word Similarity and Synonym identification between sense-based representations.

80 multiple-choice synonym questions with four choices per question. The second one, introduced by Turney (2001), provide a set of questions extracted, in this case, from the synonym questions of the ESL (English as a Second Language). Similarly to TOEFL, it is composed by 50 multiple-choice synonym questions with four choices per question.

**Word Analogy.** For word analogy we experimented with the SemEval-2012 task on Measuring Degrees of Relational Similarity (Jurgens et al., 2012). The task provides a dataset comprising 79 graded word relations, 10 of which are used for training and the rest for test. The task includes two different evaluations, one based on Spearman correlation and a second one based on MaxDiff score (Louviere, 1991).

### 5.2 Corpora and Training Details

**Corpora.** We compare several sense-annotated corpora: SemCor (Miller et al., 1994) is a manually sense-tagged corpus created by the Word-Net project team at Princeton University. The corpus is a subset of the English Brown Corpus and comprises around 360,000 words, providing annotations for more than 200K content words. The *Semantically Enriched Wikipedia* (Raganato et al., 2016, SEW) is a automatically-constructed corpus which exploits the hyperlink structure of Wikipedia and the wide-coverage sense inventory of BabelNet. Finally Scozzafava et al. (2015, BabelWiki) presented a sense-annotated corpus of the English and Italian Wikipedia, annotated automatically with Babelfy (Moro et al., 2014), a tool for Word Sense Disambiguation on the Babel-Net semantic network. We unify the sense annotations of all corpora used to learn our models into a single sense inventory. To this end we choose the last available version of BabelNet (3.7). BabelNet is both a multilingual encyclopedic dictionary, with lexicographic and encyclopedic coverage of terms, and a semantic network which connects concepts and named entities in a network of semantic relations. The resource is a merger of multiple lexical semantic resources such as Word-Net and Wikipedia.

### 5.3 Hyperparameter selection

For chosing the hyperparameters, we trained LSTMEmbed on a raw corpus[6]. We evaluated the resulting word representations on WS353, one of the word similarity datasets included in our experiments:

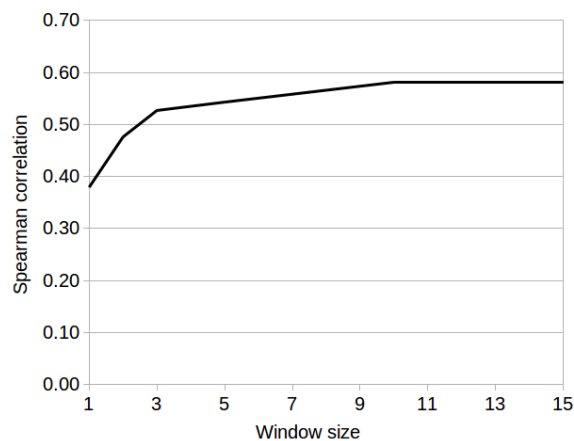

Figure 3: Effect of window size variation against Spearman correlation in WS353.

We varied the window size between 1 and 15 and we averaged the results across datasets. As in visible in Figure 3, we found that taking a window size longer than 3 words was approximately equivalent across experiments. For our experiments we used a windows size of 10 as that is the recommended window size for word2vec.

### 5.4 Learning embeddings

LSTMEmbed was built on the Keras[7] library using Theano (Theano Development Team, 2016)

---
[6] http://mattmahoney.net/dc/text8.zip
[7] https://keras.io

as backend. Keras is a neural networks library written in Python and an output of the ONEIROS project (Open-ended Neuro-Electronic Intelligent Robot Operating System). We trained our models with a Nvidia Titan X Pascal GPU on a computer equipped with an eight core Intel Core i7-3820 running at 3.60GHz with 64GB or RAM.

## 6 Results

### 6.1 Corpus Selection

In Figure 4 we can see the progression of the Spearman correlation while our model learns a larger portion of the training corpus. We can see that the BabelWiki corpus achieves better representations than SEW.

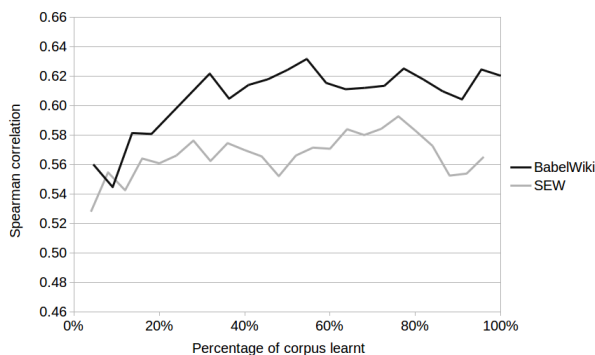

Figure 4: Spearman correlation in WS353 against learning progression on BabelWiki and SEW.

### 6.2 Word Similarity with BabelWiki

In Table 3 we show the results of our model compared with other approaches in the tasks of word similarity and synonym identification. The representations were all trained on the same sense-annotated corpus, i.e. BabelWiki. For a fair comparison we trained representations using the same configuration across algorithms: 50-dimension embeddings, window size of 10 words, and a vocabulary of 1M most frequent tokens. For handling frequent words, we discarded the 1000 most frequent tokens. We chose 50 as the LSTM dimensionality, the same dimension than our trained embeddings. We set the batch size in 128 and the training was set only for one epoch. For both word2vec and GloVe, the remaining parameters were chosen following the default configurations. For word2vec we use the Skip-gram with negative sampling set on 10, sub-sampling of frequent words set to $10^{-3}$. For GloVe we set the maximum iterations in 15, and the `x_max` in 10.

| Model | Dim | WSSim | RG65 | MEN | SimLex |
|---|---|---|---|---|---|
| SW2V | 300 | **0.710** | 0.740 | **0.760** | **0.470** |
| AutoExtend | 300 | - | - | 0.750 | 0.450 |
| SensEmbed | 400 | - | - | 0.700 | 0.390 |
| LSTMEmbed | 50 | **0.710** | 0.719 | 0.690 | 0.456 |
| LSTMEmbed$_{SW}$ | 50 | 0.626 | **0.785** | 0.644 | 0.450 |

Table 4: Spearman correlation performance of LSTMEmbed compared with alternative approaches.

The LSTMEmbed architecture obtains slightly but consistently better performance across task. The LSTMEmbed$_{SW}$ extension is consistently worse, with the exception of the RG65 dataset, but is the only approach that offers a joint space of words and sense. Contrary to our expectations, our approach does not appear to be competitive in the task on word analogy. This may be due to the fact that LSTMEmbed is a much more complex model than word2vec and GloVe. The relations in our model might be represented by a higher degree relationship.

### 6.3 Comparison with other Systems

We compare the performance of LSTMEmbed against alternative approaches able to obtain sense embeddings: SW2V, AutoExtend and SENSEMBED. Table 4 shows that our approach, although providing the embeddings with smallest dimensionality, achieves competitive results against larger embeddings from the other approaches.

### 6.4 Word Similarity with SemCor

In Table 5 we show word similarity experiments when training our model with the SemCor corpus. We performed two classes of experiments: 1) taking the SemCor corpus with all its sense annotations and 2) using a raw version of it, e.g. excluding all the annotations. The training was perform using a batch size of 32 and taking into account all the different tokens, 91349 tokens for the sense-annotated SemCor and 46797 for the runs of the raw version of SemCor–. In addition to the model trained with word2vec, we include a model, identified as LSTM, which is identical to LSTMEmbed but usese softmax as output layer as regular language models. As we can see, training with sense-annotated data outperforms the configurations based only on raw text. The LSTM model is competitive in the experiments considering only

| Model | Sense | RG65 | WS353 | WSRel | WSSim | YP130 | MEN | SCWS | SimLex |
|---|---|---|---|---|---|---|---|---|---|
| word2vec | - | 0.288 | -0.009 | 0.027 | -0.009 | 0.490 | 0.166 | 0.270 | 0.218 |
|  | ✓ | 0.397 | 0.158 | **0.125** | 0.201 | 0.268 | **0.305** | 0.325 | 0.225 |
| LSTM | - | 0.358 | 0.077 | 0.018 | 0.125 | 0.563 | 0.145 | 0.189 | 0.238 |
|  | ✓ | 0.462 | 0.071 | -0.020 | 0.128 | 0.532 | 0.108 | 0.292 | 0.247 |
| LSTMEmbed | - | 0.382 | 0.067 | 0.007 | 0.122 | 0.492 | 0.180 | 0.332 | 0.309 |
|  | ✓ | 0.434 | **0.175** | 0.102 | **0.211** | **0.650** | 0.229 | **0.351** | **0.275** |
| LSTMEmbed$_{SW}$ | - | **0.508** | 0.081 | -0.017 | 0.164 | 0.566 | 0.168 | 0.322 | 0.249 |
|  | ✓ | 0.445 | 0.125 | 0.009 | 0.210 | 0.508 | 0.214 | 0.350 | 0.269 |

Table 5: Spearman correlation for Word Similarity trained with SemCor.

|  | Emb | Dim | WS353 |
|---|---|---|---|
| word2vec | - | - | 0.488 |
| GloVe | - | - | 0.557 |
| LSTMEmbed | word2vec | 50 | 0.573 |
|  | word2ve + retro | 50 | 0.569 |
|  | GoogleNews | 300 | 0.574 |
|  | SensEmbed | 400 | **0.612** |

Table 6: LSTMEmbed with different pretrained embeddings compared with word2vec and GloVe.

raw text but unlike the other configurations, it does not seems to take advantage of the sense annotated data. We hypothesize that the increment of the vocabulary, with the corresponding increment of parameters to learn, is responsible for this behavior. Additionally, the learning time of this configuration was substantially larger than the time spent by LSTMEmbed, which is the reason why we did not include it in the experiments with large corpora. Similarly to the experiments on the BabelWiki corpus, LSTMEmbed provides good results, while the LSTMEmbed$_{SW}$ extension provides less performing results with the exception of the RG65 dataset.

### 6.5 Improving Representations via richer Embeddings

In this section we study how we can inject semantic information through the set of pretrained embeddings. Our assumption is that richer embeddings should enhance the representation delivered by our model. In Table 6 we show the behavior of the LSTMEmbed architecture in the task of word similarity with the WS353 dataset. We perform our experiments in a reduced set, i.e. a 10% sample, of our chosen sense-annotated corpus. We compared four sets of pretrained embeddings. The first set is a 50-dimension embeddings, trained with word2vec Skip-gram with the default configuration. The second set consists of the same vectors, retrofitted with PPDB using the default configuration. The third set is the well-known GoogleNews set of pretrained embeddings dataset[8]. Finally, we tested our model with the pretrained embeddings of SensEmbed[9]. As we can see, using richer pretrained embeddings improves the resulting representations given by our model. All the representations reach better results compared with the representations from word2vec and GloVe trained on the same corpus. The sense embeddings from SensEmbed, a priori the richest set of pretrained embeddings, achieved in fact the best performance.

## 7 Conclusions

In this paper we introduced LSTMEmbed, a new model based on a bidirectional LSTM for learning embeddings of words and senses. We draw four main findings: First, we have shown that our approach is able to learn word embeddings which consistently outperform the representations given by classical algorithms like word2vec and GloVe on the same training data. Second, the introduction of an output layer which predicts pre-trained embeddings allow us to inject more semantic information while speed up the training. Third, we have shown that using semantically richer embeddings as enriches the resulting representations- And finally, our sense-based representations also outperform previous representations based on word2vec and GloVe, consistently with the results shown for the word-based representations.

---

[8] https://goo.gl/p4RXac
[9] http://lcl.uniroma1.it/sensembed/

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
