# Peer review of "LSTMEmbed: a Lexical and SemanTic Model of Embeddings with a bidirectional LSTM"

_ACL 2017 — decision unknown_

[Official Review · Reviewer 1 · rating 2 · confidence 3]
soundness 4 · originality 4 · clarity 4 · impact 3 · substance 3 · appropriateness 5 · meaningful comparison 4 · presentation format Poster

- Strengths:

1. The presentation of the paper, up until the final few sections, is excellent
and the paper reads very well at the start. The paper has a clear structure and
the argumentation is, for the most part, good.
2. The paper addresses an important problem by attempting to incorporate word
order information into word (and sense) embeddings and the proposed solution is
interesting.

- Weaknesses:

 1. Unfortunately, the results are rather inconsistent and one is not left
entirely convinced that the proposed models are better than the alternatives,
especially given the added complexity. Negative results are fine, but there is
insufficient analysis to learn from them. Moreover, no results are reported on
the word analogy task, besides being told that the proposed models were not
competitive - this could have been interesting and analyzed further.
2. Some aspects of the experimental setup were unclear or poorly motivated, for
instance w.r.t. to corpora and datasets (see details below).
3. Unfortunately, the quality of the paper deteriorates towards the end and the
reader is left a little disappointed, not only w.r.t. to the results but with
the quality of the presentation and the argumentation.

- General Discussion:

1. The authors aim "to learn representations for both words and senses in a
shared emerging space". This is only done in the LSTMEmbed_SW version, which
rather consisently performs worse than the alternatives. In any case, what is
the motivation for learning representations for words and senses in a shared
semantic space? This is not entirely clear and never really discussed in the
paper.
2. The motivation for, or intuition behind, predicting pre-trained embeddings
is not explicitly stated. Also, are the pre-trained embeddings in the
LSTMEmbed_SW model representations for words or senses, or is a sum of these
used again? If different alternatives are possible, which setup is used in the
experiments?
3. The importance of learning sense embeddings is well recognized and also
stressed by the authors. Unfortunately, however, it seems that these are never
really evaluated; if they are, this remains unclear. Most or all of the word
similarity datasets considers words independent of context.
4. What is the size of the training corpora? For instance, using different
proportions of BabelWiki and SEW is shown in Figure 4; however, the comparison
is somewhat problematic if the sizes are substantially different. The size of
SemCor is moreover really small and one would typically not use such a small
corpus for learning embeddings with, e.g., word2vec. If the proposed models
favor small corpora, this should be stated and evaluated.
5. Some of the test sets are not independent, i.e. WS353, WSSim and WSRel,
which makes comparisons problematic, in this case giving three "wins" as
opposed to one.
6. The proposed models are said to be faster to train by using pre-trained
embeddings in the output layer. However, no evidence to support this claim is
provided. This would strengthen the paper.
7. Table 4: why not use the same dimensionality for a fair(er) comparison?
8. A section on synonym identification is missing under similarity measurement
that would describe how the multiple-choice task is approached.
9. A reference to Table 2 is missing.
10. There is no description of any training for the word analogy task, which is
mentioned when describing the corresponding dataset.